**Data Availability Statement:** Data are located at: https://doi.org/10.7910/DVN/WGSNWT wickramasinghe, nilanka, 2023, "The association

# The association between symptoms of gastroesophageal reflux disease and perceived stress: A countrywide study of Sri Lanka

Nilanka Wickramasinghe[1☯]*, Ahthavann Thuraisingham[2‡], Achini Jayalath[2‡], Dakshitha Wickramasinghe[3‡], Nandadeva Samarasekara[3‡], Etsuro Yazaki[4‡], Niranga Manjuri Devanarayana[5☯]

1 Department of Physiology, Faculty of Medicine, University of Colombo, Colombo, Sri Lanka, 2 Ministry of Health, Colombo, Sri Lanka, 3 Department of Surgery, Faculty of Medicine, University of Colombo, Colombo, Sri Lanka, 4 Gastrointestinal Physiology Unit, Barts and The London School of Medicine, London, United Kingdom, 5 Department of Physiology, Faculty of Medicine, University of Kelaniya, Kelaniya, Sri Lanka

☯ These authors contributed equally to this work.
‡ AT, AJ, DW, NS and EY also contributed equally to this work.
* nilanka@physiol.cmb.ac.lk

## Abstract

### Background/Aims

Stress is a known associated factor for gastroesophageal reflux disease (GERD). However, the dynamics between stress and GERD are not fully studied, especially in Sri Lanka. Our objective was to assess it.

### Methods

For this cross-sectional descriptive study, 1200 individuals (age ranged 18–70 years, mean 42.7 years [SD 14.4 years], 46.1% males), were recruited using stratified random cluster sampling from all 25 districts of Sri Lanka. An interviewer-administered questionnaire, which included a country-validated GERD symptom screening tool, and the Perceived Stress Scale (PSS), was used to assess GERD symptoms and stress. Probable GERD was defined as those having heartburn and/ or regurgitation at least once per week which is on par with globally accepted criteria. Those who did not fulfill these criteria were considered as controls.

### Results

PSS score was higher in those with probable GERD (mean 13.75 [standard deviation (SD) 6.87]) than in controls (mean 10.93 [SD 6.80]), ($p$ <0.001, Mann-Whitney U test). The adjusted odds ratio for GERD symptoms was 1.96 times higher (95% confidence interval 1.50–2.55) in the moderate to high-stress level compared to the low-stress level partici-pants. PSS score correlated significantly with the GERD screening tool score (R 0.242, $p$ <0.001). Heartburn, regurgitation, chest pain, cough, and burping were significantly frequent

between symptoms of Gastroesophageal Reflux Disease and stress: A countrywide study of Sri Lanka", https://doi.org/10.7910/DVN/WGSNWT, Harvard Dataverse, V1, UNF:6: MUMqzMNc1gcPWqZzivvsiA== [fileUNF].

**Funding:** NW UGC/VC/DRIC/PG2019(1)/CMB/01 University Grants Commision Sri Lanka https://www.ugc.ac.lk/ The funders had no role in study design, data collection and analysis, decision to publish, or preparation of the manuscript. NW Small grants University of Colombo AP/3/2/2020/SG/11 https://cmb.ac.lk/ The funders had no role in study design, data collection and analysis, decision to publish, or preparation of the manuscript.

**Competing interests:** The authors have declared that no competing interests exist.

in those with moderate to high-stress levels (p <0.001). Those with higher stress scores were more likely to use acid-lowering drugs ($p = 0.006$).

## Conclusions

Individuals exposed to higher levels of stress are more likely to have GERD symptoms. Therefore, stress reduction should be an important part of GERD symptom management.

## Introduction

Gastroesophageal reflux disease (GERD) is a disease condition where gastric contents effortlessly enter back into the esophagus, leading to unpleasant symptoms such as heartburn, regurgitation, etc. It is a common condition with an estimated global prevalence of 13.98% and a wide variance in prevalence in different countries worldwide [1]. GERD severely affects the quality of life of its sufferers, leading to poor productivity, reduced sleep, and stress [2]. If left untreated, this condition could lead to complications such as esophagitis, stricture formation, etc. [3, 4].

GERD is shown to be triggered by many risk factors, including obesity, smoking, etc., and the pathophysiology of this condition is very complex [5]. Mental stress too is a well-recognized and important risk factor for GERD symptoms in patients [6]. Stress could also reduce the effectiveness of treatments for GERD patients [7]. Furthermore, stress can be a precedent for functional dyspepsia and heartburn, which can emulate the symptoms of GERD without the underlying pathophysiology of GERD [8]. Functional heartburn being misdiagnosed for GERD is one of the reasons for treatment resistance in patients with GERD symptoms [8]. If so, anxiolytic medications can play a major role in treating patients with GERD symptoms [9].

While stress is a known associated factor for gastroesophageal reflux disease (GERD), the pattern in which stress affects the different symptoms of GERD, and the dynamics between stress and GERD are not fully researched or understood. Knowing these dynamics will help physicians pick up patients who suffer from stress-related functional dyspepsia and heartburn, or GERD patients whose symptoms are induced and aggravated by stress, thus allowing effective management of them. Furthermore, how much of an association stress has with GERD has not yet been studied in Sri Lanka. Thus, the objective of this research is to try to fill these gaps in knowledge.

## Materials and methods

### Data collection

Adult Sri Lankans (aged 18 to 70 years) who were not bed-bound or wheelchair-bound and who had not undergone gastrointestinal surgeries were recruited in this cross-sectional study conducted across all 25 districts of Sri Lanka. A sample size of 1200 was calculated using standard statistical methods (for a 0.05 level of precision, an expected prevalence of 50% (as there are no previous estimates of GERD in Sri Lanka), an allowance of 2 for a design effect of cluster sampling, and an anticipated non-response of 5%).

The sampling frame was census data of the year 2012, which was the latest census done in Sri Lanka.

We used stratified random cluster sampling to avoid bias. A cluster was defined as 30 individuals from a "Grama Niladhari" division, which is the smallest administrative unit in Sri

Lanka. Clusters were randomly chosen from each district, keeping in mind the probability proportionate to the population. Further, random selection was used to identify the participants from each cluster. The study was conducted during a three-month period from May to July in the year 2021.

Of the subjects approached, 41 (3.3%) were unable or refused consent to participate in the research. Age and sex-matched additional subjects from the same cluster, were recruited to target the sample size of 1200.

## Scales used

An interviewer-administered questionnaire that incorporated a country-specific validated questionnaire for GERD symptoms [10] and the Perceived Level of Stress (PSS) scale [11] was used in data collection. No information was obtained that could identify individual participants during or after data collection. The questionnaires were available in all three official languages of Sri Lanka, namely Sinhala, Tamil, and English.

The PSS scale is one of the most widely accepted and reliable measuring tools for perceived stress. It has been widely used in studies done in Sri Lanka and has shown acceptable composite reliability [12]. The score includes 10 items that question the user's feelings or mental state in the past month. The answers are chosen through a five-point Likert scale, while the scoring system categorizes those with low, moderate, and high-stress levels (score ranges from 0 to 13, 14 to 26, and 27 to 40, respectively) [11].

A GERD screening tool, which is the only validated tool for Sri Lanka, will assess the frequency and severity of seven GERD symptoms (namely heartburn, reflux, chest pain, bloating, dysphagia, cough, and belching). The frequency is scored from 1 to 5 (never, monthly symptoms, weekly symptoms, twice or more weekly symptoms, daily symptoms), while the severity is scored from 1 to 4 (no disability, mild, moderate, and severe). The patients are categorized as probable GERD patients if the composite GERD score is greater than 12.5 [10].

The Sinhala translations of the GERD screening tool [10] and the PSS which were available were used [13]. Translation and cultural adaptation were carried out for the Tamil versions of both, using the World Health Organization's recommended five-step process [14].

The participants were questioned on whether they used any type of proton pump inhibitor (using commonly used generic and trade names to recall), at least once, during the past three months. They were also questioned on whether the use of PPIs for heartburn, during the past three months, had "no GERD symptom relief" or "achieved GERD symptom relief."

## Definitions used

The questionnaire for GERD assesses the frequency and severity of seven GERD symptoms during the previous month to give a composite score [10]. However, to keep on par with international studies and internationally used definitions, probable GERD was defined as those having heartburn and/ or regurgitation at least once a week [1]. Unfortunately, without gold standard investigations such as pH impedance monitoring of the esophagus, our GERD categorization can also include other conditions such as functional heartburn, etc., which can mimic GERD.

## Statistical analysis

The data analysis was done using SPSS 28. There was no missing data due to the interviewer-based questionnaire. Statistical methods deemed the study population to have a non-parametric distribution. The Chi-square test was used for nominal data, the Mann-Whitney test was used for ratio data, and backward logistic regression was used to assess the independent

association between factors [15]. The tests used are incorporated in the results section, in the figures, and in the tables. A two-sided *p* less than 0.05 was considered as significant.

### Ethical approval

Ethical clearance was obtained from the Ethics Review Committee of the Faculty of Medicine, University of Colombo, and each of the nine 'Regional Director of Health Services' offices representing the nine provinces of Sri Lanka.

## Results

The study consisted of 1200 participants (age mean 42.7 years, SD 14.4 years, 46.1% males). The prevalence of GERD in Sri Lanka was 25.3% (42.1% males; 57.9% female; *p*-value = 0.111). Those who did not fulfill the GERD definition (n = 896) were considered controls.

According to the PSS score, the population was categorized as those with low-stress levels (n = 723) and those with moderate to high-stress levels (n = 477).

### Stress and GERD

Stress levels were found to be significantly higher in those with probable GERD compared to controls, as seen in Table 1. Compared to those with the lowest perceived stress level, the adjusted odds ratio (OR) for GERD symptoms in those with both moderate and high-stress levels was 1.957 times higher (95% CI = 1.504–2.547). Compared to those with the lowest perceived stress level, the GERD prevalence was also higher in those with moderate to high-stress levels (*p* <0.001).

Different demographic and health-related factors associated with stress, found by comparing probable GERD subjects with controls, are indicated in Table 2.

When the score obtained for GERD was correlated with the PSS score for the total population, there was a significant correlation in the total population (R 0.242, *p*<0.001). When the same correlation was done for the subgroups of probable GERD and controls, as seen in Fig 1, a higher correlation was found for those with GERD, (R 0.256 and 0.137, respectively), though results were statistically significant (*p* <0.001) for both GERD and controls.

When the stress level was stratified as low, moderate, and high, the mean GERD scores were 13.47 (SD 11.87), 18.20 (SD 17.27), and 28.33 (SD 32.59), respectively (Kruskal Wallis test, 39.63, DF = 2, *p* <0.001). There were statistically significant differences in the GERD score, between low and medium (*p*<0.001) and low and high (*p* = 0.005) stress levels but not between medium and high-stress levels.

**Table 1. Comparison of stress scores between probable GERD and controls.**

| | Probable GERD (*n* = 304) | Controls (*n* = 896) | *p*-value |
|---|---|---|---|
| PSS score *mean (SD)* | 13.75(6.87) | 10.93(6.80) | <0.001[a] |
| Stress level (*n*, %) | | | |
| Low | 146(48%) | 577(64.4%) | <0.001[b] |
| Moderate to severe | 158(52%) | 319(35.6%) | |
| Stress-induced exacerbation of symptoms (*n*, %) | 127(41.8%) | 89(9.9%) | <0.001[b] |

[a] Mann-Whitney U test

[b] Chi-square

PSS- perceived Stress scale

**Table 2. Analysis of factors that affect stress amongst probable GERD and controls.**

| | *Probable GERD* | | | *Control* | | |
|---|---|---|---|---|---|---|
| | **Moderate to severe stress (n = 158)** | **Low stress (n = 146)** | *p-value* | **Moderate to severe stress (n = 577)** | **Low stress (n = 319)** | *p-value* |
| Female gender (*n*, %) | 85 (53.8%) | 91 (62.3%) | 0.163[a] | 166 (52%) | 305 (52.9%) | 0.834[a] |
| Age (years) *mean (SD)* | 44.1 (13.7) | 40.8 (13.0) | 0.048 [b] | 40.4 (14.3) | 44.1 (14.7) | <0.001[b] |
| Hypertension (*n*, %) | 30 (19%) | 32 (21.9%) | 0.570[a] | 42 (13.2%) | 92 (15.9%) | 0.283[a] |
| Diabetes mellitus (*n*, %) | 27 (17.1%) | 25 (17.1%) | 1.000[a] | 35 (11%) | 72 (12.5%) | 0.521[a] |
| Current smoker (*n*, %) | 36 (22.8%) | 31 (21.2%) | 0.783[a] | 64 (20.1%) | 91 (15.8%) | 0.117[a] |
| Inadequate exercise [c] (n, %) | 140 (88.6%) | 123 (84.2%) | 0.314[a] | 263 (82.4%) | 464 (80.4%) | 0.477[a] |
| Coffee intake (*n*, %) | 55 (34.8%) | 46 (31.5%) | 0.545[a] | 121 (37.9%) | 206 (35.7%) | 0.515[a] |
| Alcohol intake (*n*, %) | 39 (24.7%) | 37 (25.3%) | 0.895[a] | 70 (21.9%) | 111 (19.2%) | 0.340[a] |
| Marital status | | | | | | |
| Married (*n*, %) | 120 (75.9%) | 128 (87.7%) | 0.011[a] | 228 (71.5%) | 478 (82.8%) | <0.001[a] |
| Single (*n*, %) | 38 (24.1%) | 18 (12.3%) | | 91 (28.5%) | 99 (17.2%) | |
| Abdominal obesity[d] (*n*, %) | 102 (64.6%) | 103 (70.5%) | 0.273[a] | 197 (61.8%) | 366 (63.4%) | 0.665[a] |
| Below Grade 11 education (*n*, %) | 84 (53.2%) | 82 (56.2%) | 0.645[a] | 160 (50.2%) | 335 (58.1%) | 0.025[a] |
| Low income <50 000Rs (*n*, %) | 98 (62%) | 91 (62.3%) | 1.000[a] | 187 (58.6%) | 358 (62%) | 0.318[a] |
| Skipping breakfast (*n*, %) | 62 (39.2%) | 51 (34.9%) | 0.477[a] | 79 (24.8%) | 128 (22.2%) | 0.408[a] |
| Reduced/interrupted sleep (*n*, %) | 71 (44.9%) | 60 (41.1%) | 0.567[a] | 102 (32%) | 162 (28.1%) | 0.222[a] |

[a] Pearson Chi-square

[b] Mann-Whitney U test

[c] Inadequate physical activity (<600 MET minutes/week)

[d] Abdominal obesity if abdominal circumference is more than 90 cm in males and 80 cm in females according to cut-offs for Asians

### Stress and GERD symptoms

Those with moderate to high-stress levels were experiencing a significantly higher number of different GERD-related symptoms than those with low-stress levels, with a mean score of 1.09 and an SD of 1.48 in those with low-stress levels, whereas those with moderate to high-stress levels had a mean score of 1.64 and an SD of 1.72 (a significance of $p < 0.001$ was calculated using the Mann-Whitney test).

Table 3 compares the frequency and severity scores of seven GERD-related symptoms among low and moderate to high-stress groups.

We also noted that patients with chest pain reported higher stress levels than those without, though the result was not statistically significant. (48.4% and 57.9%, Chi-square test, $p = 0.124$).

Another observation made when comparing using the Pearson Chi-square test was that those with moderate to high levels of stress versus those with no or low levels of stress were more likely to have used a proton pump inhibitor for the past three months at the point of questioning (23.5% versus 17.4%, $p = 0.006$). However, there was also a significant association between stress levels, and the lack of symptom alleviation in those who were on medication for heartburn ($p < 0.001$), as 55.6% of those with high to moderate stress had no relief on medication for heartburn, whereas the percentage without symptom relief for medication was 44.4% of those with low-stress levels.

### Discussion

This is the first study reporting a possible association between emotional stress and GERD symptoms in Sri Lanka. In this study, the PSS score was higher in those with frequent GERD

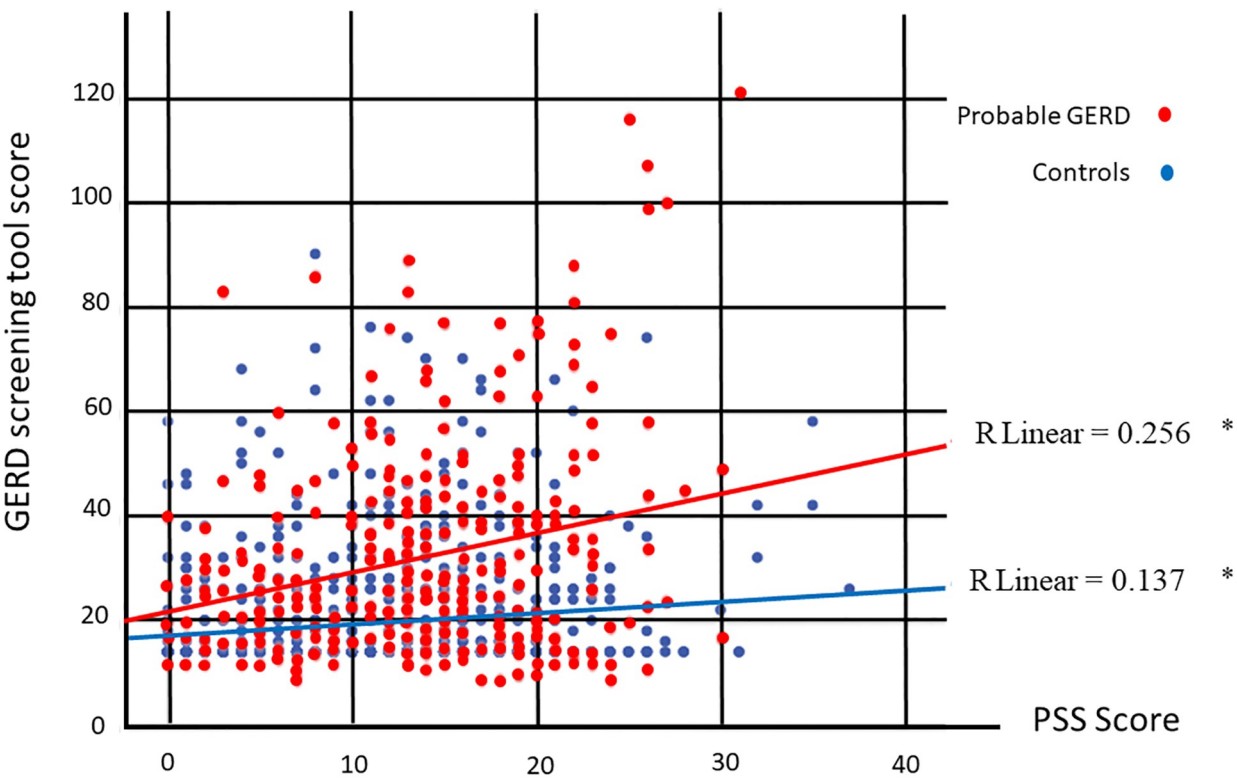

**Fig 1. Simple scatter with fit line comparing GERD screening tool score with PSS score of the probable GERD and controls.** * significance <0.001 using Pearson correlation.

**Table 3. Comparison of frequency and severity of individual GERD symptoms between low-stress level and moderate to high-level stress participants.**

| | Total population (n = 1200) | | | Probable GERD (n = 304) | | |
|---|---|---|---|---|---|---|
| | Low-stress levels (n = 723) | Moderate to high-stress levels (n = 477) | p-value[a] | Low-stress levels (n = 146) | Moderate to high-stress levels (n = 158) | p-value[a] |
| | mean (SD) | mean (SD) | | mean (SD) | mean (SD) | |
| Heartburn frequency score | 1.6 (1.1) | 1.93 (1.3) | <0.001 | 3.5 (1.3) | 3.4 (1.3) | 0.789 |
| Heartburn disability score | 1.4 (0.7) | 1.6 (1.0) | <0.001 | 2.3 (0.9) | 2.6 (1.0) | 0.010 |
| Regurgitation frequency score | 1.4 (0.9) | 1.7 (1.2) | <0.001 | 2.7 (1.5) | 2.8 (1.5) | 0.783 |
| Regurgitation disability score | 1.2 (0.5) | 1.4 (0.8) | <0.001 | 1.8 (0.8) | 2.0 (1.0) | 0.039 |
| Chest pain frequency score | 1.2 (0.7) | 1.4 (1.0) | <0.001 | 1.7 (1.2) | 1.0 (1.4) | 0.091 |
| Chest pain disability score | 1.2 (0.6) | 1.4 (0.8) | <0.001 | 1.5 (0.9) | 1.8 (1.1) | 0.014 |
| Abdominal bloating frequency score | 1.6 (1.1) | 1.7 (1.2) | 0.051 | 2.7 (1.6) | 2.4 (1.5) | 0.124 |
| Abdominal bloating disability score | 1.2 (0.6) | 1.4 (0.7) | 0.004 | 1.7 (0.9) | 1.7 (1.0) | 0.883 |
| Dysphagia frequency score | 1.1 (0.56) | 1.1 (0.6) | 0.182 | 1.2 (0.9) | 1.3 (1.0) | 0.341 |
| Dysphagia disability score | 1.0 (0.28) | 1.1 (0.4) | 0.048 | 1.1 (0.5) | 1.2 (0.6) | 0.331 |
| Cough frequency score | 1.1 (0.5) | 1.3 (0.9) | <0.001 | 1.3 (0.9) | 1.5 (1.2) | 0.175 |
| Cough disability score | 1.1 (0.3) | 1.1 (0.5) | 0.018 | 1.2 (0.5) | 1.3 (0.7) | 0.231 |
| Burping frequency score | 1.4 (1.1) | 1.7 (1.3) | <0.001 | 2.3 (1.6) | 2.3 (1.6) | 0.564 |
| Burping disability score | 1.1 (0.3) | 1.2 (0.5) | <0.001 | 1.2 (0.5) | 1.4 (0.7) | 0.066 |

[a] Mann-Whitney U test

symptoms (heartburn and regurgitation) than in those without such frequent symptoms. In addition, the PSS score correlated significantly with the GERD symptom score (obtained from the GERD screening tool), suggesting the significant impact of stress on symptom severity. The severity and/or frequency of individual symptoms such as heartburn, regurgitation, chest pain, bloating, and burping were significantly increased in those with moderate to high-stress levels than those with low-stress levels. Furthermore, those with higher stress scores were more likely to use more acid-lowering drugs, which further shows the impact of stress.

Despite "GERD" being a common topic, its exact prevalence in different regions of the world, including South Asia, is not well established. The unavailability of the globally accepted and validated diagnostic tool for the diagnosis of GERD, which can be used for large epidemiological studies, is probably the main reason for this. The 24-hour combined pH impedance testing that is currently considered the gold standard [16] is time-consuming, invasive, expensive, and only available in specialized gastroenterology units. Therefore, this investigation cannot be used to measure the community prevalence of GERD. Studies worldwide, given this diagnostic dilemma, have resorted to using different symptoms-based criteria for GERD, which have resulted in the large variations in prevalence reported [1]. Currently, the most widely accepted definition used for probable GERD is "having heartburn and/or regurgitation at least once a week" [1]. A systematic review conducted in 2020 using this definition reported a global prevalence of 13.98% after analyzing 102 studies [1]. Using the same definition, this countrywide study found the prevalence of probable GERD as 25.3% in Sri Lanka, which is higher than the global prevalence reported in the systematic review. This is even higher than the previously reported highest prevalence in Turkey (22.4%) and many Western countries, including the United Kingdom and the United States of America, and Asian countries, including China and India [1, 17, 18].

The pathophysiology of GERD is not clear, and many mechanisms have been suggested. They include malfunctioning lower esophageal sphincter (LES), transient LES relaxations, impaired acid clearance, and increased intragastric pressure [5]. There are many possible risk factors for GERD, such as emotional stress, obesity, smoking, familial preponderance, pregnancy, hiatal hernia, impaired gastric motility, medications, etc. [19–22].

Prolonged GERD results in either erosive esophagitis with mucosal damage or non-erosive GERD, which is mainly due to increased sensitivity of the esophageal mucosa to acid. Acid exposure and associated inflammation stimulate nociceptors in the epithelial intercellular space. They in turn activate esophageal vagal and spinal afferents, which send sensory information to the cortex, causing the sensation of heartburn. In a normal individual, most of the reflux events cause minimal activation of the nociceptors and are not perceived [23, 24]. Prolonged acid exposure, mucosal inflammation, and hypersensitivity of the nociceptors result in significant heartburn and sometimes trigger sustained esophageal contractions, causing atypical chest pain [24].

The main differential diagnosis for GERD is functional dyspepsia or functional heartburn, where the subject has no reflux evidence either by ambulatory pH monitoring or endoscopy but still complains of heartburn. They are termed functional heartburn sufferers [8].

Studies done worldwide have shown that stress increases GERD symptoms [6, 25, 26]. According to longitudinal studies main stressful events associated with GERD symptoms include divorce, the death of a spouse, miscarriage, and severe automobile accidents [7]. Symptoms of GERD are also noted to be higher in patients with phobias, obsessive-compulsive disorders, and interpersonal sensitivities [27].

A study conducted in Saudi Arabia found that GERD symptoms were significantly ($p < 0.05$) associated with high perceived stress (OR = 1.30, 95% CI: 1.01–1.44), which is

compatible with our results [28]. Furthermore, another study reported that 60% of GERD patients perceived increased symptoms during stressful times [29]. This is compatible with our results, which show aggravation of symptoms in 41.8% of subjects during periods of stress.

A previous study reported a significantly lower level of anxiety, in diagnosed patients with GERD compared to functional heartburn [30]. A prospective study in Sweden also found that anxiety was a risk factor for developing functional dyspepsia, but not GERD symptoms [31]. According to these studies, increased levels of anxiety are more likely to be associated with GERD symptoms due to functional heartburn than true GERD. However, since our study did not do pH impedance testing, we cannot show the difference in stress levels between subjects with functional disorders and those objectively diagnosed with GERD.

Furthermore, studies have found that patients who had symptomatic erosive GERD on endoscopy had higher levels of anxiety, as opposed to asymptomatic patients who were incidentally found to have erosive esophagitis [32]. It is possible that stress can increase sensitivity and lower the threshold of pain, making the distinction between symptomatic and asymptomatic GERD.

Of all the symptoms of GERD, chest pain is the one most associated with stress. GERD patients with chest pain have more anxiety and stress than GERD patients without chest pain [33–35]. In the current study, considering the total population of 1200, the frequency and severity of heartburn, regurgitation, chest pain, burping, and cough, as well as the severity of bloating were significantly higher in individuals exposed to high or moderate stress levels compared to low stress. However, when doing the same analysis on participants categorized as having probable GERD, only the severity of heartburn, regurgitation, and chest pain were noted to be significant at a $p$-value <0.05. These symptoms are known to increase with increasing mental stress [7, 36–40]. While studies have also shown that stress can increase the severity of dysphagia [37], we did not observe a significant difference in the perception of dysphagia between participants with probable GERD and controls.

Stress is defined as any physical (real) or psychological (perceived) threat to an organism's homeostasis [41]. Thus, in psychological terms, stress is the term for feeling emotional strain or pressure [42]. The response to such a psychological threat can usually become maladaptive due to frequent and chronic stresses such as financial or interpersonal problems [41]. Individuals with such maladaptive responses may be predisposed to disease conditions in multiple organs, including the GI (gastrointestinal) tract, through various pathophysiological mechanisms [43]. Recent studies have demonstrated that psychological stress also affects the perception of nociceptors [44].

Increased anxiety and stress levels are postulated to affect the GI tract and esophagus in many ways, subsequently triggering true GERD, as well as other conditions such as reflux hypersensitivity and functional heartburn, which give rise to similar symptoms. Induction of these symptoms can be modulated through regulators such as the brain-gut axis, the enteric nervous system, the neuroendocrine system of the gut, the immune system of the gut, and gut microbiota. These result in the alteration of gut motility and secretions, as well as peripheral and central sensitization resulting in reflux hypersensitivity.

Stress can cause dysfunction in the central regulation of the GI tract via the brain-gut axis, by altering secretions, mucosal barrier functions, permeability, visceral sensation, and motility [45]. This axis can be activated through bi-directional neurological as well as hormonal signaling [46, 47]. Dysfunction of the brain-gut axis associated with stress can lead to many GI symptoms, including GERD.

Visceral hypersensitivity of the esophagus due to peripheral and central sensitization [24] is a well-known cause of GERD symptoms. Visceral hypersensitivity is present in patients with

true GERD, as well as in those with reflux sensitivity and functional heartburn [48]. The most quoted mechanism by which stress is thought to cause visceral hypersensitivity and GERD symptoms is by increasing esophageal permeability and dilated intercellular spaces (DIS), which in turn activate nociceptors [44, 43]. In addition, stress can modulate pain and other nociceptive stimuli at the central level, by acting on visceral brain areas such as the amygdala, periaqueductal region, anterior cingulate gyrus, and prefrontal cortex [25].

A study done in 2022 showed that the gut microbial diversity was significantly different between patients with GERD and healthy controls [49]. The exact reason for this difference and its impact on the generation of symptoms are complex and poorly understood. It may be due to gut dysbacteriosis and may even affect the drug efficacy [49]. Emotional stress has a significant impact on the microbiota of the gut and has an impact on GI symptoms [47]. It has been shown that even short-term stress can change the gut microbiota profile [50].

Stress is shown to have an impact on GI motor functions. Young *et al*. recorded esophageal manometry in participants exposed to stresses in the form of noise disturbances and problem-solving tasks and reported, increased stress-induced LES relaxations [51]. In addition, smooth muscle dysfunction is thought to cause delayed gastric emptying and reduced gastric motility, in functional GI disorders [52]. Similar stress-induced impairments in gastric emptying and motility can lead to increased intragastric pressure, thereby increasing the risk of GERD.

Increased acidity can give rise to more GERD-related symptoms through multiple mechanisms. It is commonly perceived that stress is a major trigger for increased acid secretion. However, studies assessing the relationship between stress and gastric acid secretion have shown contradictory results, showing stress-induced increased acid secretion in some individuals while others have a reduction in acid secretion in response to stress [53, 54].

Previous studies have suggested a relationship between stress and GI immune functions. Stress-induced activation of mast cells in the esophagus is likely to sensitize the nociceptors, as well as cause dysfunction of local neural networks and the autonomic nervous system [44, 55].

There is also another disease condition termed functional heartburn, whereby the patient complains of heartburn, but pH studies show only normal amounts of physiological reflux with no association between heartburn and these refluxes [56]. It is thought to be caused by many pathophysiological pathways, such as increased esophageal perceptions, neuronal dysfunction, or greater access of nociceptors to acid [43, 57].

Mucosal barrier dysfunction and increased permeability of the gastrointestinal mucosa are also said to be part of the pathophysiology of Functional Gastrointestinal Disorders (FGIDs) [58]. The hypothalamic-pituitary-adrenal axis can also play a role in the release of corticotropin-releasing factors, causing visceral hypersensitivity and gut dysmotility in FGID patients [59]. An immune dysfunction caused by the release of mast cells and eosinophils is also thought to lead to pain in patients with FGID [60, 61].

Not only GERD symptoms but even other GI symptoms such as bloating, discomfort, diarrhea, constipation, and extra-intestinal symptoms such as neck aches and back aches are also said to be higher in those with higher perceived stress levels [62, 63]. This could be a feature of increased somatization associated with psychological stress.

Some behaviors induced by stress are well-recognized risk factors for GERD. Previous studies have reported an association between GERD and female gender, current smoking, obesity, sleep interruption, low income, and low education status [64, 65]. The current study also identified several relevant associated factors with symptoms of GERD, such as younger age, being single, and low education status (Table 2).

An inverse relationship has been reported between age and high stress [64]. In our study, GERD symptoms were more common among younger participants, and high-stress levels may have partially contributed to this.

Studies have shown that anxiety, depression, and other mood disorders increase the odds of developing GERD-related symptoms significantly [6, 66]. In this community-based study, we did not assess such psychiatric conditions due to practical difficulties.

When it comes to the treatment of GERD, the reduction of stress plays an important role. This could be in the form of relaxation training and stress management, which have been shown to significantly lower reflux symptom ratings and total esophageal acid exposure in GERD patients [67].

Many treatment modalities that modify stress are already recommended (e.g., anxiolytics) as effective treatments for GERD symptoms, especially in functional heartburn [8].

Our study found that those with high levels of stress were more likely to be on PPIs, but had, significantly lower symptomatic relief from anti-acid medications. Similar findings have been reported in studies done worldwide [68].

This study has several strengths. The participants were selected from stratified random sampling throughout the entire country from all 25 districts and are representative of the distributions of age, sex, ethnicities, and religions in Sri Lanka. Secondly, we used a sound methodology, with country-validated structured questionnaires with culturally validated translations.

Of the limitations, first, due to the cross-sectional design of our study, it was not possible to perform endoscopy and pH impedance studies, and therefore, we could not confirm whether GERD symptoms reported by patients are due to true GERD. Secondly, we asked the participants to recall their symptoms for the past month, which could cause recall or responder bias. However, we used a standard and validated screening tool to assess symptoms and a widely used and worldwide accepted criterion for the diagnosis of probable GERD, which enabled us to compare our results with studies conducted globally.

In conclusion, in this island-wide community-based study, we found a significant association between stress and GERD symptoms such as heartburn and regurgitation. In addition, those who are exposed to high levels of stress are more likely to use long-term acid-lowering medications such as PPI, but their response to such medications is significantly lower than that of those not exposed to such high levels of stress. Therefore, alleviation of stress is an important component in the prevention of GERD and its management. Medications lowering anxiety and stress, such as anxiolytics and antidepressants, and psychological and behavioral therapies such as mindfulness meditation, CBT, and yoga, are likely effective management options for GERD symptoms, and well-designed therapeutic trials should be conducted to assess their exact therapeutic value.

## Author Contributions

**Conceptualization:** Nilanka Wickramasinghe, Niranga Manjuri Devanarayana.

**Data curation:** Nilanka Wickramasinghe, Ahthavann Thuraisingham, Achini Jayalath, Dakshitha Wickramasinghe.

**Formal analysis:** Nilanka Wickramasinghe, Ahthavann Thuraisingham, Achini Jayalath, Dakshitha Wickramasinghe.

**Funding acquisition:** Nilanka Wickramasinghe, Niranga Manjuri Devanarayana.

**Investigation:** Nilanka Wickramasinghe, Ahthavann Thuraisingham, Achini Jayalath.

**Methodology:** Nilanka Wickramasinghe, Niranga Manjuri Devanarayana.

**Project administration:** Nilanka Wickramasinghe, Ahthavann Thuraisingham, Achini Jayalath.

**Supervision:** Nandadeva Samarasekara, Etsuro Yazaki.

**Writing – original draft:** Nilanka Wickramasinghe, Niranga Manjuri Devanarayana.

**Writing – review & editing:** Nandadeva Samarasekara, Etsuro Yazaki.

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
