## [Decision Letter · Decision Letter 0]

23 Jun 2023

PONE-D-23-13957The association between Gastroesophageal Reflux Disease and stress: A countrywide study of Sri LankaPLOS ONE

Dear Dr. wickramasinghe,

Thank you for submitting your manuscript to PLOS ONE. After careful consideration, we feel that it has merit but does not fully meet PLOS ONE’s publication criteria as it currently stands. Therefore, we invite you to submit a revised version of the manuscript that addresses the points raised during the review process.

We look forward to receiving your revised manuscript.

Kind regards,

Karthik Gangu

Academic Editor

PLOS ONE

Journal Requirements:

3. Please upload a new copy of Figure 1 as the detail is not clear. Please follow the link for more information: 

https://blogs.plos.org/plos/2019/06/looking-good-tips-for-creating-your-plos-figures-graphics/

https://blogs.plos.org/plos/2019/06/looking-good-tips-for-creating-your-plos-figures-graphics/

Reviewers' comments:

Reviewer's Responses to Questions

**Comments to the Author**

1. Is the manuscript technically sound, and do the data support the conclusions?

Reviewer #1: Yes

Reviewer #2: Yes

2. Has the statistical analysis been performed appropriately and rigorously? 

Reviewer #1: Yes

Reviewer #2: Yes

3. Have the authors made all data underlying the findings in their manuscript fully available?

Reviewer #1: No

Reviewer #2: Yes

4. Is the manuscript presented in an intelligible fashion and written in standard English?

Reviewer #1: Yes

Reviewer #2: Yes

5. Review Comments to the Author

Reviewer #1: Dear Authors,

It was a pleasure to read your work titled “The association between Gastroesophageal Reflux Disease and stress: A countrywide study of Sri Lanka”.

Overall, a good study with clinical implications. Few concerns

Major

As pointed out by the authors, this is a study on association of various symptoms to the presence/degree of stress. Not necessarily GERD as the study participants did not undergo any formal testing to evaluate for the presence of true evidence of pathological acid reflux. Many of these symptoms can be seen in GERD by are also shared by other etiologies such as functional dyspepsia, functional heartburn, reflux hypersensitivity to name a few. As such, I recommend modifying the title and discussion to reflect this.

Minor

Overall, there are multiple long, complex sentences throughout this manuscript. Some have grammatical errors too. This forces the reader stop and retrace the sentence to understand the discussion. I would recommend revising to make the flow smooth and easy on the reader.

Results section is a little confusing and feels scattered

Lines 163 to 166. Unclear what the authors are conveying here.

Lines 187 to 189. Do the authors mean "patients with chest pain reported higher stress compared to those without chest pain, but did not reach statistical significance". This sentence is difficult to follow.

Discussion

Line 250. "This pattern is observed in our results as well as seen in Table 3." I would elaborate this a little and clarify.

Lines 273 to 276. The idea is conveyed but the sentence should be revised.

"Increased anxiety and stress levels are postulated to affect the GI tract and esophagus in many ways, subsequently triggering true GERD, as well as other conditions which rise to similar symptoms such as reflux hypersensitivity and functional heartburn"

Line 316 introduces the abbreviation - FGID without expanding/explaining this. Please expand.

Line 326 This sentence is incomplete

"The current study also identified some associated factors such as younger age, being single, and a low education status" -- Associated with what should be clarified.

Lines 330 to 334. I think this can be made more succinct.

Reviewer #2: Overall, a well written manuscript on a relevant topic. here are my comments -

1. One fundamental issue with the definition of GERD is based on symptomatology and may encompass several patients with functional dyspepsia and other esophageal motility disorder. This brings the idea of stress directly related to GERD into question. Further detailed explanation of association of symptomatology and perceived stress level would be helpful.

2. Provide further explanation regarding lack of symptomatic improvement with GERD medications (what medications were tried and for what duration) in patients with high levels of stress.

6. PLOS authors have the option to publish the peer review history of their article (what does this mean?). If published, this will include your full peer review and any attached files.

Reviewer #1: **Yes: **Harishankar Gopakumar, MD, FACP

Reviewer #2: No

---

## [Author Response · Author response to Decision Letter 0]

4 Aug 2023

9th July 9, 2023 

Dear Editors and Reviwers,

Thank you very much for the comprehensive review of the paper. It helped to greatly enhance the research idea. We have done the required revisions. This document shows the point-by-point responses to the corrections and questions posed by both editors and reviewers. My responses are given in red text for ease of reading. Please let me know if there are any mistakes I can revise if not done so far.

Sincerely

Nilanka Wickramasinghe 

Done

We have uploaded the data set to a repository as requested.

Please see data set https://doi.org/10.7910/DVN/WGSNWT available at HARVARD Dataverse

https://dataverse.harvard.edu/file.xhtml?fileId=7236399&version=DRAFT

Dataset Citation

wickramasinghe, nilanka, 2023, "The association between symptoms of Gastroesophageal Reflux Disease and stress: A countrywide study of Sri Lanka", https://doi.org/10.7910/DVN/WGSNWT, Harvard Dataverse, DRAFT VERSION, UNF:6:MUMqzMNc1gcPWqZzivvsiA== [fileUNF]

File Citation

wickramasinghe, nilanka, 2023, "The association between symptoms of Gastroesophageal Reflux Disease and stress: A countrywide study of Sri Lanka", https://doi.org/10.7910/DVN/WGSNWT, Harvard Dataverse, DRAFT VERSION; DataforstressinGERDpaper.tab [fileName], UNF:6:MUMqzMNc1gcPWqZzivvsiA== [fileUNF] 

3. Please upload a new copy of Figure 1 as the detail is not clear. Please follow the link for more information: 

https://blogs.plos.org/plos/2019/06/looking-good-tips-for-creating-your-plos-figures-graphics/

https://blogs.plos.org/plos/2019/06/looking-good-tips-for-creating-your-plos-figures-graphics/

Done

Reviewer's Responses to Questions

Comments to the Author

1. Is the manuscript technically sound, and do the data support the conclusions?

Reviewer #1: Yes

Reviewer #2: Yes

2. Has the statistical analysis been performed appropriately and rigorously?

Reviewer #1: Yes

Reviewer #2: Yes

3. Have the authors made all data underlying the findings in their manuscript fully available?

Reviewer #1: No - data shared

Reviewer #2: Yes

4. Is the manuscript presented in an intelligible fashion and written in standard English?

Reviewer #1: Yes

Reviewer #2: Yes

5. Review Comments to the Author

Reviewer #1: Dear Authors,

It was a pleasure to read your work titled “The association between Gastroesophageal Reflux Disease and stress: A countrywide study of Sri Lanka”.

Overall, a good study with clinical implications. Few concerns

Major

As pointed out by the authors, this is a study on association of various symptoms to the presence/degree of stress. Not necessarily GERD as the study participants did not undergo any formal testing to evaluate for the presence of true evidence of pathological acid reflux. Many of these symptoms can be seen in GERD by are also shared by other etiologies such as functional dyspepsia, functional heartburn, reflux hypersensitivity to name a few. As such, I recommend modifying the title and discussion to reflect this.

Revised title as recommended.

“The association between symptoms of Gastroesophageal Reflux Disease and perceived stress: A countrywide study of Sri Lanka”

Edits made for discussion section as well.

Minor

o Overall, there are multiple long, complex sentences throughout this manuscript. Some have grammatical errors too. This forces the reader stop and retrace the sentence to understand the discussion. I would recommend revising to make the flow smooth and easy on the reader.

We have revised accordingly with use of expert help and software to reduce complex sentences and grammar errors.

o Results section is a little confusing and feels scattered

We have rearranged the results section with advice given by the reviewer.

o Lines 163 to 166. Unclear what the authors are conveying here.

The lines 163 to 166 were rewritten as given below.

“When the score obtained for GERD was correlated with the PSS score for the total population, there was a significant correlation in the total population (R 0.242, p<0.001). When the same correlation was done for the subgroups of GERD and controls as seen in Figure 1, a higher correlation was found for those with GERD, (R 0.256 and 0.137 respectively), though results were statistically significant (p<0.001) for both GERD and controls.”

o Lines 187 to 189. Do the authors mean "patients with chest pain reported higher stress compared to those without chest pain, but did not reach statistical significance". This sentence is difficult to follow.

Yes. We have rewritten as the reviewer has suggested.

“Patients with chest pain reported higher stress levels than those without, though the result was not statistically significant. (48.4% and 57.9%, Chi-square test, p= 0.124).”

o Discussion

Line 250. "This pattern is observed in our results as well as seen in Table 3." I would elaborate this a little and clarify.

It’s a line erronousely added. We changed from line 241 to 251 it to fit the reviewer’s comment at the very beginning regarding the GERD definition. 

“A similar study conducted in Saudi Arabia found that GERD symptoms were significantly (p < 0.05) associated with high perceived stress (OR = 1.30, 95% CI: 1.01–1.44) which is compatible with our results. [31] Furthermore, another study reported that 60% of GERD patients perceived increased symptoms during stressful times. [32] This is compatible with our results which show aggravation of symptoms in 41.8% of subjects during periods of stress.

A previous study has reported a significantly lower level of anxiety, in diagnosed patients with GERD compared to functional heartburn [33] A prospective study in Sweden also found that anxiety was a risk factor for developing functional dyspepsia, but not GERD symptoms. [34] According to these studies, increased levels of anxiety are more likely to be associated with GERD symptoms due to functional heartburn than true GERD. However, since our study did not do pH impedance testing we cannot show the difference in stress levels between subjects with functional disorders and those objectively diagnosed with GERD.”

o Lines 273 to 276. The idea is conveyed but the sentence should be revised.

"Increased anxiety and stress levels are postulated to affect the GI tract and esophagus in many ways, subsequently triggering true GERD, as well as other conditions which rise to similar symptoms such as reflux hypersensitivity and functional heartburn"

Thank you for revising and making the sentence more understandable. We added the sentence revised by the reviewer.

o Line 316 introduces the abbreviation - FGID without expanding/explaining this. Please expand.

Corrected. 

Mucosal barrier dysfunction and increased permeability of the gastrointestinal mucosa are also said to be part of the pathophysiology for Functional Gastrointestinal Disorders (FGIDs).

o Line 326 This sentence is incomplete

"The current study also identified some associated factors such as younger age, being single, and a low education status" -- Associated with what should be clarified.

Corrected.

The current study also identified some associated factors such as younger age, being single, and a low education status with symptoms of GERD. (Table 2). 

o Lines 330 to 334. I think this can be made more succinct.

We have edited the lines as given below

“Studies have shown that anxiety, depression, and other mood disorders increase the odds of developing GERD-related symptoms significantly.[71] [71] In this community-based study, we did not assess such psychiatric conditions due to practical difficulties.”

Reviewer #2: Overall, a well written manuscript on a relevant topic. here are my comments -

1. One fundamental issue with the definition of GERD is based on symptomatology and may encompass several patients with functional dyspepsia and other esophageal motility disorder. This brings the idea of stress directly related to GERD into question. Further detailed explanation of association of symptomatology and perceived stress level would be helpful.

The discussion is amended and edited according to the reviewer comments

We have also changed the title and discussion related to “association of stress to GERD symptoms” as opposed to just “GERD”

2. Provide further explanation regarding lack of symptomatic improvement with GERD medications (what medications were tried and for what duration) in patients with high levels of stress.

The methodology section was revised to include the following.

“The participants were questioned on whether they used any type of proton pump inhibitors (using commonly used generic and trade names to recall), at least once, during the past three months.

They were also questioned on whether the use of PPIs for heartburn, during the past three months, had “no GERD symptom relief” or “achieved GERD symptom relief.”

---

## [Decision Letter · Decision Letter 1]

16 Oct 2023

PONE-D-23-13957R1The association between Gastroesophageal Reflux Disease and stress: A countrywide study of Sri LankaPLOS ONE

Dear Dr. wickramasinghe,

Thank you for submitting your manuscript to PLOS ONE. After careful consideration, we feel that it has merit but does not fully meet PLOS ONE’s publication criteria as it currently stands. Therefore, we invite you to submit a revised version of the manuscript that addresses the points raised during the review process.

We look forward to receiving your revised manuscript.

Kind regards,

Dong Keon Yon, MD, FACAAI, FAAAAI

Academic Editor

PLOS ONE

Journal Requirements:

Additional Editor Comments:

Please see minor comments from me and the reviewers.

#1. The Chi-square test was used for nominal data, the Mann-Whitney test was used for Wickramasinghe ratio data, and backward logistic regression was used to assess the independent association between factors. -> Please cite the statistical guideline (DOI: https://doi.org/10.54724/lc.2022.e1).

#2. A two-sided P less than 0.05 considered significance -> Please add this sentence.

#3. Please receive English Editing Service.

Reviewers' comments:

Reviewer's Responses to Questions

**Comments to the Author**

1. If the authors have adequately addressed your comments raised in a previous round of review and you feel that this manuscript is now acceptable for publication, you may indicate that here to bypass the “Comments to the Author” section, enter your conflict of interest statement in the “Confidential to Editor” section, and submit your "Accept" recommendation.

Reviewer #1: All comments have been addressed

2. Is the manuscript technically sound, and do the data support the conclusions?

Reviewer #1: Yes

3. Has the statistical analysis been performed appropriately and rigorously? 

Reviewer #1: Yes

4. Have the authors made all data underlying the findings in their manuscript fully available?

Reviewer #1: Yes

5. Is the manuscript presented in an intelligible fashion and written in standard English?

Reviewer #1: Yes

6. Review Comments to the Author

Reviewer #1: Thank you for incorporating the recommended changes. I believe it is now good for publication with a few minor edits if possible.

1. Under results section (Lines 38 to 39) "The adjusted odds ratio for GERD symptoms was 1.96 times higher (95% confidence interval 1.50-2.55) between moderate to high-stress level and low-stress level participants.

I feel this sentence should be restructured to "The adjusted odds ratio for GERD symptoms was 1.96 times higher (95% confidence interval 1.50-2.55) in the moderate to high-stress level compared to the low-stress level participants.

2. Introduction section lines 60-62.

"GERD is shown to be triggered by many risk factors, including obesity, smoking, etc., and the pathophysiology of this condition is very complex. Conversely, mental stress is a well-

recognized and important risk factor for GERD symptoms in patients"

Comment - Why say "conversely" here? It is not a contradictory statement to the preceding one. I recommend amending this.

7. PLOS authors have the option to publish the peer review history of their article (what does this mean?). If published, this will include your full peer review and any attached files.

Reviewer #1: No

---

## [Author Response · Author response to Decision Letter 1]

23 Oct 2023

Journal Requirements:

We have revised and edited the references and referencing list to be up-to-date, and have removed some references.

Additional Editor Comments:

Please see minor comments from me and the reviewers.

#1. The Chi-square test was used for nominal data, the Mann-Whitney test was used for Wickramasinghe ratio data, and backward logistic regression was used to assess the independent association between factors. -> Please cite the statistical guideline (DOI: https://doi.org/10.54724/lc.2022.e1).

We have cited the references and advised.

#2. A two-sided P less than 0.05 considered significance -> Please add this sentence.

We have added this sentence as advised. “A two-sided p less than 0.05 was considered as significant.”

#3. Please receive English Editing Service.

We used Grammarly software and an expert in English to go through the paper.

Reviewer #1: Thank you for incorporating the recommended changes. I believe it is now good for publication with a few minor edits if possible.

1. Under results section (Lines 38 to 39) "The adjusted odds ratio for GERD symptoms was 1.96 times higher (95% confidence interval 1.50-2.55) between moderate to high-stress level and low-stress level participants.

I feel this sentence should be restructured to "The adjusted odds ratio for GERD symptoms was 1.96 times higher (95% confidence interval 1.50-2.55) in the moderate to high-stress level compared to the low-stress level participants.

Thank you for the comments. We have restructured the lines as advised.

2. Introduction section lines 60-62.

"GERD is shown to be triggered by many risk factors, including obesity, smoking, etc., and the pathophysiology of this condition is very complex. Conversely, mental stress is a well-

recognized and important risk factor for GERD symptoms in patients"

Comment - Why say "conversely" here? It is not a contradictory statement to the preceding one. I recommend amending this.

We have amended the lines as advised. “Mental stress too is a well-

recognized and important risk factor for GERD symptoms in patients.”

---

## [Editor Report · Decision Letter 2]

26 Oct 2023

The association between symptoms of Gastroesophageal Reflux Disease and perceived stress; A country wide study of Sri Lanka

PONE-D-23-13957R2

Dear Dr. wickramasinghe,

We’re pleased to inform you that your manuscript has been judged scientifically suitable for publication and will be formally accepted for publication once it meets all outstanding technical requirements.

Kind regards,

Dong Keon Yon, MD, FACAAI, FAAAAI

Academic Editor

PLOS ONE

Additional Editor Comments (optional):

This is an excellent paper.
---

## [Editor Report · Acceptance letter]

31 Oct 2023

PONE-D-23-13957R2 

The association between symptoms of Gastroesophageal Reflux Disease and perceived stress: A countrywide study of Sri Lanka 

Dear Dr. Wickramasinghe:

I'm pleased to inform you that your manuscript has been deemed suitable for publication in PLOS ONE. Congratulations! Your manuscript is now with our production department. 

Kind regards, 

on behalf of

Dr. Dong Keon Yon 

Academic Editor

PLOS ONE